# Uncertainty Evaluation of Soil Heavy Metal(loid) Pollution and Health Risk in Hunan Province: A Geographic Detector with Monte Carlo Simulation

**DOI:** 10.3390/toxics11121006

**Published:** 2023-12-08

**Authors:** Baoyi Zhang, Yingcai Su, Syed Yasir Ali Shah, Lifang Wang

**Affiliations:** 1Key Laboratory of Metallogenic Prediction of Nonferrous Metals and Geological Environment Monitoring (Ministry of Education), School of Geosciences and Info-Physics, Central South University, Changsha 410083, China; zhangbaoyi@csu.edu.cn (B.Z.); 215012143@csu.edu.cn (Y.S.); 235008003@csu.edu.cn (S.Y.A.S.); 2Department of Surveying and Mapping Geography, Hunan Vocational College of Engineering, Changsha 410151, China

**Keywords:** nemerow index, non-carcinogenic risk, carcinogenic risk, factor detector, interaction detector, uncertainty propagation

## Abstract

Research on soil heavy metal(loid) pollution and health risk assessment is extensive, but a notable gap exists in systematically examining uncertainty in this process. We employ the Nemerow index, the health risk assessment model, and the geographic detector model (GDM) to analyze soil heavy metal(loid) pollution, assess health risks, and identify driving factors in Hunan Province, China. Furthermore, the Monte Carlo simulation (MCS) method is utilized to quantitatively evaluate the uncertainties associated with the sampling point positions, model parameters, and classification boundaries of the driving factors in these processes. The experimental findings reveal the following key insights: (1) Regions with high levels of heavy metal(loid) pollution, accompanied by low uncertainty, are identified in Chenzhou and Hengyang Cities in Hunan Province. (2) Arsenic (As) and chromium (Cr) are identified as the primary contributors to health risks. (3) The GDM results highlight strong nonlinear enhanced interactions among lithology and other factors. (4) The input GDM factors, such as temperature, river distance, and gross domestic product (GDP), show high uncertainty on the influencing degree of soil heavy metal(loid) pollution. This study thoroughly assesses high heavy metal(loid) pollution in Hunan Province, China, emphasizing uncertainty and offering a scientific foundation for land management and pollution remediation.

## 1. Introduction

Heavy metal(loid)s possess characteristics of bioaccumulation, persistence, and toxicity. Excessive accumulation of heavy metal(loid) in soil can deteriorate soil quality, posing a threat to the stability of ecosystems. Additionally, heavy metal(loid)s can enter the human body through oral ingestion, inhalation, and skin contact, ultimately damaging human health [1]. The sources of soil heavy metal(loid) pollution can be classified into natural and anthropogenic factors [2]. Natural sources mainly arise from the presence of heavy metal(loid)s in the parent rock materials and soil formation processes [3,4,5], while anthropogenic sources primarily include emissions from transportation activities, industrial discharges into water and air, and agricultural wastewater containing excessive pesticides and fertilizers [6]. Due to the rapid socio-economic development in China, soil heavy metal(loid) pollution has become an increasingly severe environmental issue. Therefore, investigating the spatial and temporal distribution patterns and sources of soil heavy metal(loid) pollution is crucial for controlling and mitigating this issue.

Currently, most research on soil heavy metal(loid) pollution assessment is based on pollution indexes [7,8]. Common indexes include the potential ecological risk index [9], the geo-accumulation index [10], the and Nemerow index [11]. Health risk assessment aims to link environmental pollution with human health and quantitatively describe the degree of harm that environmental pollution poses to human health. The health risk assessment model proposed by the United States Environmental Protection Agency (USEPA) (https://www.epa.gov/, accessed on 30 September 2023) is widely used in the world [12]. Although this model is relatively practical, most of the previous studies ignored the model uncertainty. Identifying and quantifying the contributions of various pollution sources to soil heavy metal(loid) contamination is of great significance in formulating corresponding measures to reduce and control pollution sources. Common methods used for soil pollution source analysis include isotope tracing [13], multivariate statistical methods [14], positive matrix factorization (PMF) [15], and geographical information system (GIS) spatial analysis [16,17]. However, these methods can only globally classify pollution sources in a region based on existing data and experiences, which may introduce uncertainties and ignore the spatial characteristics of pollution sources and the interactions between factors.

The spatial correction between heavy metal(loid)s and environmental factors has been extensively studied. The geographic detector model (GDM), proposed by Wang et al. [18], excels in detecting spatial heterogeneity and its driving mechanism. Utilizing GDM, Huang et al. [6] evaluated the effects of eighteen environmental factors on soil heavy metal pollution in Zhangzhou City, China, covering six natural factors and twelve anthropogenic factors. Wang et al. [19] analyzed the driving forces of heavy metal distribution in different cultivated land quality subdivisions in the Yangtze River delta region with GDM. It is crucial to note that GDM necessitates the dependent variable to be numerical and the independent variables to be categorical. If the independent variables are numerical, they must be categorized, and the uncertainty arising from different classification boundaries may be further transmitted in GDM. Monte Carlo simulation (MCS), a common uncertainty assessment method, aids in evaluating the uncertainty in environmental pollution and health risk assessment processes for more reliable results [20,21,22]. MCS is widely used to assess uncertainties of exposure variables (soil ingestion rate, average body weight, exposed skin area, and skin adherence) to reflect individual differences in human health risk models. For example, Barrio-Parra et al. [23] assessed the effect of variability and uncertainty on all exposure variables. Zhou et al. [24] utilized MCS to evaluate the probabilistic health risks associated with a smelter in Hunan Province. However, measurement errors in sampling point locations and concentrations have been overlooked. Meanwhile, few studies have discussed uncertainty propagation in Nemerow soil pollution evaluation and GDM in conjunction with MCS.

To address these issues, we employ MCS to jointly analyze uncertainty propagation effects related to sampling point locations, heavy metal(loid) concentrations, exposure variables, and impact factor boundaries on soil pollution assessment, human health risk assessment, and geo-detector models. In soil pollution assessment, MCS explores uncertainty propagation from sampling positions and heavy metal(loid) concentrations on the Nemerow index, quantifying pollution level uncertainty using information entropy. For human health risk assessment, MCS investigates uncertainty propagation from heavy metal(loid) concentrations and exposure variables on the health risk index, generating cumulative probability curves. In GDM, MCS probes uncertainty propagation from impact factor category boundaries on Nemerow indices using factor detector and interaction detector models. This study enriches the theoretical methodologies for addressing uncertainty propagation in soil heavy metal(loid) pollution and health risk assessment models, offering vital support for more precise control of heavy metal(loid) pollution in Hunan Province.

## 2. Study Area and Dataset

### 2.1. Study Area

Hunan Province is in the transitional zone between the Yunnan-Guizhou Plateau and the low hills and terrains on the south side of the Yangtze River. Spanning from 108°47′ to 114°15′ east longitude and 24°38′ to 30°08′ north latitude, it encompasses elevations ranging from 21 to 2122 m. Covering an area of 21.18 × 10^4^ square kilometers, it features a continental subtropical monsoon humid climate, with an average annual temperature of 15–18 °C and an average annual precipitation of 1200–1700 mm. Hunan Province is surrounded by mountains on its east, south, and west sides, with undulating hills and hillocks in the central region and flat basin plains in the north. Centered around Dongtinghu Lake, it is traversed by four major rivers, i.e., Xiangjiang, Zijiang, Yuanjiang, and Lishui, covering approximately 96.7% of the total provincial area. Known as “the land of fish and rice”, Hunan boasts a cultivated land area of 4.1488 million hectares, representing around 3.1% of China’s total cultivated land area, with rice, ramie, and tea as the main crops.

The soil texture is diverse in Hunan province, including clay, clay loam, silty loam, sandy clay loam, and loamy sand. The strata are well developed in Hunan, and geological formations consist primarily of sedimentary rocks, e.g., sandy rocks, carbonate rocks, red rocks, and Quaternary loose deposits, covering about 57.75% of the total land area. Metamorphic rocks constitute approximately 24.99% and igneous rocks about 8.87% (Appendix A). Known as “the lands of non-ferrous metals and non-metals”, it hosts 39 metal deposits, mainly in the southeastern region, with over 140 identified mineral types.

Hunan province has a long history of non-ferrous metal mining, smelting, chemical industries, and other activities involving heavy metal(loid) discharge, making the non-ferrous metal industry a prominent pillar industry. While significantly contributing to the provincial economy and social development, these industries have also led to heavy metal(loid) pollution issues. Activities such as mechanical engineering, electronic information, new materials, lead, zinc carbide, and other industrial and mining processes result in the accumulation of heavy metal(loid)s in the soil, posing potential health risks. For instance, the cadmium-contaminated rice incident, a consequence of heavy metal(loid) contamination in agricultural land, has garnered significant attention.

### 2.2. Dataset

#### 2.2.1. Soil Sampling Points

The soil sampling points were obtained from the census data of soil heavy metal(loid) pollution in agricultural production areas of Hunan Province conducted by the Hunan Land and Resources Planning Institute (http://www.hngtghy.com/, accessed on 30 September 2023), encompassing a total of 48,811 sampling points. These data include the contents of chromium (Cr), cadmium (Cd), arsenic (As), lead (Pb), mercury (Hg), and pH value. The sampling site locations span various agricultural land scenarios: arable land (paddy and dryland), garden land, forest land, and grassland. A multi-point mixture approach was conducted using an “S” shaped or plum blossom coupled points with random sampling techniques, extracting surface soil samples at depths of 0–20 cm. These samples underwent natural air drying in the laboratory, sieving, and subsequent acid digesting, adhering to the standards outlined in HJ/T 166-2004 [25]. Determination of heavy metal(loid)s was carried out according to the method specified in GB 15618-2018 [26].

The descriptive statistics of soil heavy metal(loid) concentrations in Hunan Province are presented in Appendix A. The mean and standard deviation (SD) of pH value of the soil is 5.83 ± 0.93, with a range from 2.1 to 8.7. The mean concentrations of Cr, Cd, As, Pb, and Hg are all higher than the soil background values from Li et al. [27], with values of 1.129, 5.92, 1.37, 2.15, and 2 times the background values, respectively. This indicates that Cd pollution is more severe in the study area. Cr and As have low or near-background levels, indicating that they might originate from natural sources. There is a significant variation in heavy metal(loid) concentrations among the sampling points, and the maximum values for all five heavy metal(loid)s exceed the lowest risk screening values by 8.54, 1391.7, 189.1, 139.5, and 471.4 times, respectively, indicating the existence of severe heavy metal(loid) pollution in some areas of Hunan Province. The coefficient of variation (CV) reflects the average variation degree of each sampling point. The CV values for soil heavy metal(loid) and soil pH, from the largest to the smallest, are as follows: Hg (8.94) > Cd (5.41) > Pb (2.49) > As (2.29) > Cr (0.4) > pH (0.16). Larger CV and skewness values indicate more significant spatial variations and susceptibility to human activities [15].

The spatial distribution map of soil heavy metal(loid) concentrations in the study area was generated using the inverse distance weighted (IDW) interpolation method according to soil sampling points (Figure 1). Regions with high concentrations of Cr are primarily located in areas such as Changsha and Xiangtan Cities, possibly due to improper disposal of industrial waste from electroplating, battery, and stainless-steel industries [28]. Areas with high concentrations of Cd are mainly distributed at the junction of Huayuan and Baojing Counties, as well as along the Xiangjiang River between Changning and Hengnan Counties, with mining activities, industrial wastewater, and residue discharges being the main sources of Cd pollution. Regions with high concentrations of As and Pb show a relatively similar distribution, mainly in the southeastern part of Hunan Province, particularly concentrated in Changning County and Chenzhou City. This is consistent with Chenzhou City being a world-known non-ferrous metal center, aligning with the higher levels of As and Pb content. Areas with high concentrations of Hg are found in Yongzhou Lingling District and Baojing County in Western Hunan Province, primarily originating from coal combustion pollution [15].

#### 2.2.2. Influencing Factors

The gross domestic production (GDP), population density (PD), mining activity areas (MA), roads, and rainfall were derived from the results of the coupling evaluations of the carrying capacity of resources and environment and suitability of land spatial planning of Hunan Province conducted by the Hunan Land and Resources Planning Institute. Slope and aspect were extracted from a digital elevation model (DEM) with a spatial resolution of 90 m, obtained from NASA’s Space Shuttle Radar Terrain Mission (https://srtm.csi.cgiar.org/, accessed on 30 September 2023). The daily average temperature of Hunan meteorological stations was downloaded from the National Oceanic and Atmospheric Administration (NOAA) (https://www.noaa.gov/, accessed on 30 September 2023) and then averaged to obtain the annual average temperature. Additionally, the soil map of Hunan Province at a scale of 1:10 million was obtained from the Institute of Soil Science, Chinese Academy of Sciences (http://www.issas.ac.cn/, accessed on 30 September 2023). Considering the practical situation of Hunan Province, we selected sixteen factors that influence soil heavy metal(loid) accumulation. Among them, the ten natural factors are slope, aspect, lithology (Lith), soil type (ST), soil organic carbon (SOC), pH, water and soil erosion (WSL), temperature (TEM), precipitation (PRE) and distance to streams (DS). Additionally, six anthropogenic factors are land-use/land-cover (LULC), gross domestic product (GDP), population density (PD), mining activity areas (MA), distance to railways (DRW), and distance to roads (DR).

In this study, we applied the optimal discretization method, available in the GD package of R Studio software (https://posit.co/download/rstudio-desktop/, accessed on 30 September 2023), to categorize the distances from railways, roads, and streams and annual average temperature. For the remaining factors, we utilized prior knowledge for categorization. For example, the slope data were categorized based on the slope classification of the geomorphic detail map provided by the International Geographical Union (https://igu-online.org/, accessed on 30 September 2023). Soil texture was divided into thirteen categories according to the size of soil particles, and pH was grouped into four categories, considering specific situations with agricultural significance. Figure 2, Figure 3 and Appendix A show the specific categories of sixteen factors.

## 3. Methods

Our approach involves three main parts (Figure 4): (1) Utilizing the Nemerow index and USEPA model to reveal the distribution characteristics of Cr, Cd, As, Pb, and Hg pollution levels, as well as associated human health risks in Hunan Province. (2) Employing GDM to quantify the contributions of natural and anthropogenic risk sources to soil heavy metal(loid) pollution and investigating the interactions among different risk sources. (3) Utilizing MCS to simulate the uncertainties of sampling point locations, attributes, exposure variables, and impact factor categorial boundaries, then quantitatively calculating the uncertainty in the evaluation process. This comprehensive evaluation will contribute to more accurate soil heavy metal(loid) pollution control and prevention measures.

### 3.1. Nemerow Pollution Index with MCS

#### 3.1.1. Nemerow Pollution Index

The Nemerow index has been proven to be effective in quantifying the overall pollution level of soil heavy metal(loid)s [11]. It assesses the average pollution level of various soil contaminants and highlights the impact and significance of pollutants with the highest pollution index on environmental quality [19]. The calculations for the single-factor pollution index and the Nemerow pollution index are shown in Formula (1) and (2), respectively.
(1)$$P_{i}=\frac{C_{i}}{S_{i}}$$
where $P_{i}$ is the pollution index of a single heavy metal(loid) i; $C_{i}$ is the concentration of heavy metal(loid)s i (mg/kg); and $S_{i}$ is the evaluation standard of heavy metal(loid)s i (mg/kg) in soil. In this study, GB 15618-2018 [26] was adopted as the evaluation standard, and specified risk screening values are outlined in Appendix A.
(2)$$P_{n}=\sqrt{\frac{P_{ave}^{2}+P_{max}^{2}}{2}}$$
where $P_{n}$ is the Nemerow pollution index of heavy metal(loid)s in soil; $P_{ave}$ is the average value of the single factor index; $P_{max}$ is the maximum value of the single factor pollution index. The classification criteria for the assessment of soil heavy metal(loid) pollution are presented in Appendix A.

#### 3.1.2. Uncertainty of the Nemerow Pollution Index

MCS represents one of the most prevalent and effective approaches for characterizing uncertainty in numerous risk-related problems [29]. MCS involves several steps: (1) defining random variables of the assessment model, (2) setting distribution models for these random variables, (3) configuring simulation parameters and executing the model, and (4) analyzing the simulation outcomes. In this study, the PyMC package was employed for MCS (https://www.pymc.io/, accessed on 30 September 2023), with 5000 iterations conducted, and the results were derived from the last 3000 iterations.

The uncertainty assessment for heavy metal(loid) pollution evaluation in Hunan Province is detailed as follows: (1) MCS was utilized to model random variables, including X and Y coordinates of sampling points and the concentrations of five heavy metal(loid)s, with the variable distribution specified in Table 1. (2) Based on multiple simulation outcomes, inverse distance weighted (IDW) interpolation was employed to generate distribution maps of five heavy metal(loid)s with a 1 × 1 km grid size in Hunan Province. (3) The Nemerow pollution indices were repeatedly computed according to Equations (1) and (2), and their corresponding distribution diagrams were graded into different pollution degrees. (4) The uncertainty of pollution assessment results was quantitatively represented by information entropy.

Information entropy, originally defined by Shannon [30] to quantify uncertainty in information, involves partitioning the entire model space into regular grids with uniform pixel size. For each grid cell, if there is only one possible outcome with $P_{i}$ = 1, the entropy value is 0, indicating no uncertainty. However, the more possible M outcomes, the greater the entropy and the greater the uncertainty. The general expression for information entropy is calculated by Formula (3).
(3)$$H=-\overset{M}{\underset{i}{\sum }}P_{i}logP_{i}$$

### 3.2. Health Risk Assessment with MCS

#### 3.2.1. Health Risk Assessment

Human health risk assessment is the process of predicting the potential harmful effects of environmental pollutants on human health, divided into non-carcinogenic risk assessment and carcinogenic risk assessment. Heavy metal(loid)s in the soil mainly enter the human body through two exposure pathways: ingestion and dermal contact. According to USEPA [12,31], the average exposure dose for each pathway ($ADD_{ing}$ and $ADD_{dermal}$) can be calculated using Formulas (4) and (5). Due to behavioral and physiological differences between adults and children, this study discusses the health risks that can be generated for these two groups separately.
(4)$$ADD_{ing}=\frac{CS\times IR_{ing}\times EF\times ED}{BW\times AT}\times 10^{-6}$$
(5)$$ADD_{dermal}=\frac{CS\times AF\times SA\times ABS\times EF\times ED}{BW\times AT}\times 10^{-6}$$
where the main symbols used in the formulas are explained in Appendix A.

The non-carcinogenic risk can be expressed by the $HQ_{i}$ as the sum of exposure pathways of each heavy metal(loid) *i* and the total index (*HI*) is the sum of *HQ* of heavy metal(loid)s, calculated by Formulas (5) and (6), respectively.
(6)$$HQ_{i}=\sum \frac{ADD_{k}}{RfD_{k}}$$
(7)$$HI=\sum HQ_{i}$$
where RfD represents the non-carcinogenic average daily reference dose for heavy metal(loid)s *i* in mg·(kg·d)^−1^, k means different pathways of exposure.

The carcinogenic risk for each heavy metal(loid) ($CR_{i}$) and the total carcinogenic risk (*TCR*) can be estimated using Formulas (8) and (9), respectively.
(8)$$CR_{i}=\sum ADD_{k}\times SF_{k}$$
(9)$$TCR=\sum CR_{i}$$
where SF is the carcinogenicity slope factor in (kg·d)·mg^−1^.

#### 3.2.2. Uncertainty of Health Risk Assessment

Data uncertainty in human health risk assessment arises from environmental variations, population characteristics, and insufficient scientific understanding of parameters and variables. MCS can enhance the quality and quantity of information, thereby reducing parameter uncertainty [23]. Although the Bayesian approach demonstrates precision with small sample sizes [32,33], MCS offers wider applicability and aligns better with the extensive data used in this study. In this study, stochastic variables of pollutant concentration (*CS*), ingestion rate (*IR_ing_*), exposure duration (*ED*), skin adherence factor (*AF*), skin area exposed to soils (*SA*), and average body weight (*BW*) were chosen for simulation, and their specified values are provided in Appendix A [34,35,36].

Upon ranking health risk assessment results and categorizing them into predetermined frequency bins, a cumulative probability distribution plot was generated to depict the output. The probabilistic modeling aims to capture the uncertainty propagation from heavy metal(loid) concentrations and exposure variables on the health risk index.

### 3.3. Geographic Detector Model with MCS

#### 3.3.1. Geo-Detector

The geo-detector is a novel statistical method designed to identify spatially stratified heterogeneity and unveil the driving factors contributing to it [37]. Its core idea is that if an independent variable has a more significant influence on a dependent variable, its spatial distributions should be more similar. GDM requires categorical variables as inputs for independent variables and numerical variables for dependent variables. It consists of factor, risk, ecological, and interaction detectors. In this study, our primary focus is on the factor and interaction detectors, aiming to quantitatively analyze the contributions of natural and anthropogenic factors and their interplay in influencing soil heavy metal(loid) pollution.

The factor detector assesses the accumulated dispersion variance of each sub-region in comparison to the dispersion variance of the entire study region. The smaller the ratio, the stronger the contribution. The magnitude of contribution is indicated by the q-statistic, representing the percentage (100 × q%) of variance in the dependent variable that can be explained by the independent variable. The q-statistic ranges from 0 to 1 and is calculated by Formula (10).
(10)$$q=1-\frac{\sum_{h=1}^{L}N_{h}\sigma_{h}^{2}}{N\sigma^{2}}=1-\frac{SSW}{SST}$$
where *h* = 1, 2, 3, …, *L* represents the partition of factor X, $N_{h}$ and *N* are the number of cells in the partition *h* and the entire region, respectively, and $\sigma_{h}^{2}$ and $\sigma^{2}$ are the variances of the dependent variable *Y* values of partition *h* and the whole region, respectively. *SSW* and *SST* represent the sum of square variance and the total sum of square variance, respectively [38].

The interaction detector, by comparing the sum of the contribution of two individual attributes q(X_1_) and q(X_2_) with the contribution of the two attributes when taken together q(X_1_ ∩ X_2_), identifies whether they are independent or if their joint effect enhances or weakens the explanatory power on the dependent variable Y. The types of interaction between the two factors are presented in Appendix A.

#### 3.3.2. Uncertainty of Geo-Detector

To explore the impact of classification boundary uncertainty on the results of the GDM, this study utilized the geographic spatial domain uncertainty theory and the weight determination method proposed by Zhang et al. [39]. MCS was employed to model the uncertainty of grid classification boundaries. The simulation process includes: (1) A MCS was used to draw random numbers from a uniform distribution between 0 and 1. (2) The weights of each pixel $p_{ij}$ were calculated based on the influence from different categories within its 5 × 5 neighborhood. The calculation method is illustrated in Formulas (11)–(13). (3) Each pixel’s value was re-assigned based on the inverse distance weighted ratio within specific intervals. By employing this process, new raster classification data were obtained and used as the input for the GDM. As shown in Figure 5, $p_{ij}$ represents the central pixel, and $p_{xy}$ represents a certain pixel in the neighborhood spatial area of the central pixel $p_{ij}$, where *i* and *j* are the row and column numbers of the central pixel in the raster, and x and y are the row and column numbers within the neighborhood spatial area. Different colors represent different categories, namely C1, C2, C3, and C4.

The influence weight of the class of $p_{xy}$ on the class of $p_{ij}$ can be calculated as follows:(11)$$C_{k}=\overset{N}{\underset{n=1}{\sum }}W_{kxy}$$
(12)$$W_{kxy}=\frac{\frac{1}{D_{kxy}}}{\sum_{x=1}^{5}\sum_{y=1}^{5}\frac{1}{D_{xy}}}$$
(13)$$D_{xy}=\sqrt{{\left(X_{p_{xy}}-X_{p_{ij}}\right)}^{2}+{\left(Y_{p_{xy}}-Y_{p_{ij}}\right)}^{2}}+1$$
where $C_{k}$ represents the total weight of each category, *k* = (1, 2, 3, …, *K*), and *K* is the number of classes in the neighborhood. $W_{kxy}$ denotes the inverse distance weight proportion of unit $p_{xy}$ in the neighborhood of $p_{ij}$ that belongs to category *k*, *n* = (1, 2, 3, …, *N*), *N* is the number of pixels in the neighborhood that belong to category k, *X* and *Y* are the *X* and *Y* coordinates of $p_{xy}$, $D_{xy}$ represents the distance between $p_{xy}$ and $p_{ij}$. To avoid division by zero, a value of 1 is added after calculating the distance.

## 4. Results and Discussions

### 4.1. Soil Heavy Metal(loid) Pollution Assessment

In this study, the Nemerow pollution index ($P_{n}$) was used to analyze the level of heavy metal(loid) pollution in the study area. The results show that $P_{n}$ at the mean value of the sampling point is 1.86 and $P_{n}$ at the median value is 0.99. The value of $P_{n}$ at the sampling point was skewed to the right, indicating that there were more extreme values at the right tail and the degree of dispersion on the right side of the data mean was strong. The overall pollution level of most sampling point is in the unpolluted, warning, and low pollution levels, and some points are in the medium level and high pollution levels. Figure 6 illustrates that high pollution is predominantly concentrated in the southeastern region of Hunan Province, specifically in the northwestern area of Chenzhou, as well as in the cities of Changning County and Yongzhou Lingling District. Additionally, scattered high-pollution areas are observed in locations such as Shaoyang County, the junction between Xiangtan and Zhuzhou Cities, the boundary between Taoyuan and Anhua Counties in Yiyang City, and the vicinity of Yuanling County in Huaihua City and Huayuan County in Western Hunan Province. However, it should be noted that the distribution near the junction of Zhuzhou and Xiangtan Cities exhibits considerable variations. Nevertheless, most of the medium to high-risk areas are close to the key remediation areas of mining activities in Hunan Province.

In our study, we employed the MCS method to simulate uncertainties of the soil sampling point locations and the heavy metal(loid) concentrations. Subsequently, we recalculated the $P_{n}$ for multiple perturbated datasets. The maximum uncertainty of heavy metal(loid)s is ranked as follows: Pb > Cr > As > Hg > Cd. The regions with high uncertainty are predominantly located in the northwestern and southeastern parts of Hunan Province (Appendix A).

Based on the MCS results of the information entropy distribution of $P_{n}$, the uncertainty of soil heavy metal(loid) pollution exhibits a pattern of higher uncertainty in the northern and southern regions and lower uncertainty in the central part (Figure 7a). Moreover, it illustrates the propagation of location and attribute uncertainties of heavy metal(loid) sampling points during the evaluation process. By combining the pollution risk and uncertainty results, a control zoning map (Figure 7b) is generated. The majority of regions in Hunan Province fall into the low-risk-low-uncertainty category. Regions with low-risk-high-uncertainty and high-risk-high-uncertainty are more scattered, and government authorities should enhance monitoring efforts in such regions. Furthermore, the high-risk-low-uncertainty areas show a high level of consistency with the distribution of high pollution indices. Therefore, immediate measures are recommended to address soil pollution in Hengyang and Chenzhou Cities.

### 4.2. Probabilistic Health Risk Assessment 

To mitigate the uncertainties arising from deterministic parameters in the health risk assessment model, MCS was applied to the parameters within Formulas (3)–(8). Subsequently, cumulative distribution curves F(a) = P(x ≤ a) were plotted for the health risk indices derived from multiple MCS iterations (Figure 8). The mean non-carcinogenic risk indices (*HQ*) of adults for Cr, Cd, As, Hg, and Pb were found to be 5.6 $\times \;10^{-2}$, 5.2 $\times \;10^{-3}$, 1.7 $\times \;10^{-1}$, 2.9 $\times \;10^{-3}$, and 2.4 $\times \;10^{-2}$, respectively. An *HQ* value below 1 indicates that the specific heavy metal(loid) in the soil does not pose a non-carcinogenic health risk to the human population. Conversely, an *HQ* value exceeding 1 signifies the presence of non-carcinogenic health risks. Among these, only the non-carcinogenic risk index for As in the adult population has a 1% probability of exceeding 1. In contrast, in the children population, As, Cr, and Pb have probabilities of 81%, 21%, and 1%, respectively, of exceeding the threshold. Pb and Hg have relatively minor effects on both adults and children. The total non-carcinogenic health risk (*HI*) for the adult and child populations had probabilities of 1% and 95%, respectively, of exceeding 1. This indicates that non-carcinogenic risks are almost certainly present in groups of children and demand attention.

Carcinogenic risk (*CR*) refers to the increased incidence rate of cancer resulting from lifetime exposure to carcinogens. Generally, when *TCR* is less than 1 $\times \;10^{-6}$, the carcinogenic risk can be considered negligible. If *TCR* falls within the range of 1 $\times \;10^{-6}$ to 1 $\times \;10^{-4}$, the carcinogenic risk can be deemed acceptable. However, if *TCR* exceeds 1 $\times \;10^{-4}$, the population is at a higher risk of carcinogenic effects. Figure 9 illustrates that *CR* values of both As and Cr exceed 1 $\times \;10^{-6}$ in the adult population, with 3% and 2% probabilities of surpassing 1 $\times \;10^{-4}$, respectively. This suggests that the carcinogenic risk from As and Cr for adults is generally within an acceptable range, but there still remains a small possibility of a carcinogenic risk. For the children population, the probabilities of exceeding 1 $\times \;10^{-4}$ for As and Cr are 24% and 20%, respectively. The total carcinogenic risks (*TCR*) for adults and children have probabilities of 15% and 72% of surpassing 1 $\times \;10^{-4}$, respectively. These results indicate a significant carcinogenic potential that demands attention from relevant authorities.

The results indicate that the main metals posing a high carcinogenic risk to both children and adults are As, followed by Cr, and finally, Cd. Previous studies have also confirmed that individuals are more susceptible to exposure to arsenic in the soil [28,40,41]. Zhang et al. [42] analyzed probabilistic health risks from rice ingestion in Hunan’s Zijiang River basin, finding that the 50th and 95th quantiles of As surpass the threshold for carcinogenic and non-carcinogenic risks in both children and adults. Children, due to their behavioral characteristics, are more sensitive to soil pollution. This finding is due to the strong carcinogenicity of Cr under skin exposure [43]. The CR values primarily depend on the toxicity, concentration, and mobility of heavy metal(loid)s [40,44,45]. Therefore, the higher CR values for As and Cd are mainly attributed to their relatively higher carcinogenicity slope factor (SF) values and their bioavailability [24]. Future investigations should explore exposure variable differences in oral ingestion, inhalation via nose and mouth, and dermal contact across arable land, garden land, forest land, and grassland in various scenarios.

### 4.3. Driving Factors on Heavy Metal(loid) Accumulation

#### 4.3.1. Dominant Individual Factor

The GDM serves as a valuable tool for investigating the controlling factors of soil heavy metal(loid) spatial patterns and their interactions. Based on the factor detector in GDM, Table 2 provides the quantification of the impact degree of each factor on the accumulation of heavy metal(loid) elements. The results indicate that, except for population density (PD), the result significance levels (*p* values) for all other factors are less than 0.01, signifying that the sixteen selected factors significantly influence the accumulation of heavy metal(loid) elements. Among the natural factors, DS (0.07149) and aspect (0.07105) exert the most substantial impact on the spatial distribution of soil heavy metal(loid) pollution, followed by PRE (0.07105) and slope (0.07104). Within the anthropogenic factors, DRW (0.7114), MA (0.07103), and LULC (0.07102) have the most significant influence.

Cr and As show low or near-background levels, suggesting a potential natural origin. DS, aspect, slope, and PRE are identified as the primary natural influencing factors. Shi et al. [46] pointed out that the spatial distribution of As in urban soil is mainly governed by topographical factors resulting from weathering and subsequent pedogenesis. Precipitation affects soil heavy metal(loid)s mainly in the following aspects: (1) The transport and output of heavy metal(loid)s in the soil are influenced by precipitation; (2) Precipitation may cause the diffusion of atmospheric heavy metal(loid)s to the surface [38].

Anthropogenic factors, e.g., DRW, MA, LULC, and others, play a crucial role in heavy metal(loid) pollution distribution. The findings of Yang et al. [47] suggest that industrial activity had a significant influence on the levels of Cd and Pb. Actually, non-metallic mineral products, chemical raw materials, and chemical manufacturing are also the main pillar industries in Hunan Province, and the smelting of non-ferrous metals, combustion of fossil fuels, such as coal, and the treatment of chemical waste are among the major contributors to soil pollution. There are a total of 10,160 mines in Hunan Province, with the highest numbers located in Shaoyang (1342), Hengyang (1056), and Chenzhou (1005) Cities. These findings are consistent with the areas previously identified as having high soil pollution. Li et al. [48] found a significant positive correlation between Cd and the proportion of urban area, while Huang et al. [6] demonstrated that grassland has the largest factor detector q value for Pb. According to the investigation, grassland in Hunan Province is mainly distributed in Yongzhou, Chenzhou, Shaoyang, and Hengyang, aligning with high-concentration areas of Pb. Previous studies have also indicated that heavy metal(loid)s, especially Pb, are supplemented to the soil near the railroads [49,50]. The consistency of these results with high coefficients of variation and severalfold concentrations of background for Cd and Pb further confirms the susceptibility of these elements to human activities and reaffirms the reliability of the GDM.

The moderate coefficient of variation for Hg indicates that it is influenced by both human factors and other natural sources. According to related studies, significant sources of mercury accumulation in soil include industrial activities such as fossil fuel combustion, non-ferrous metal smelting, and cement production, which lead to its enrichment in soil through atmospheric processes [36].

#### 4.3.2. Interaction between Factors

In general, the spatial distribution of heavy metal(loid)s in soil results from the combined effects of multiple factors in a complex external environment. This study utilizes the interaction detector in GDM to explore the types of interactions between two factors and their joint effects on the spatial differentiation of soil heavy metal(loid)s. The q12 values of the joint interactions between the two factors range from 0.014 to 0.15, which are generally higher than the q values of single factors (0.00129–0.07149). Among the joint interactions observed, the combined effect of Lith and DS yields the largest q12 value, with most joint interactions exhibiting high q12 values. Lith, ST, WSL, pH, and SOC showed significant interactive effects when combined with other factors, indicating that their contributions are more pronounced when they interact with other factors rather than acting individually.

The results from the interaction detector indicate that the combined effects of multiple factors are generally greater than the individual effects of single factors. Particularly, except for PD, the interaction of Lith with other factors has been identified as nonlinearly enhanced, while the interactions of other driving factors are categorized as bivariate enhanced. Figure 10 provides an illustrative representation of the interaction types among several groups of factors.

#### 4.3.3. Uncertainty of GDM

Harmon et al. [51] pointed out that model uncertainty includes prediction and selection uncertainties. Model prediction uncertainty arises during the process of transforming measured values into other variables of interest, and it can be estimated using MCSs. In this study, we used MCS simulated grid data as input variables and measured the standard deviation of 100 runs of the GDM model to quantify model prediction uncertainty. Through the analysis of standard deviation from 100 GDM results, we found that the factors with the largest standard deviation in the factor detector q values are TEM (1.57 $\times \;10^{-2}$), GDP (1.98 $\times \;10^{-3}$), and DS (7.23 $\times \;10^{-4}$). Figure 11 illustrates the distribution characteristics of these three factors. The wide range of TEM and DS q values contributes to their large standard deviation, while GDP exhibits a relatively uniform distribution within its range. For the interaction detector q values, the factors with the largest standard deviation are PD and TEM (1.32 $\times \;10^{-2}$), TEM and Lith (1.26 $\times \;10^{-2}$), and DS and Lith (4.79 $\times \;10^{-3}$). The three-dimensional scatter plot of q1-q2-q12 (Figure 12) further confirms that TEM and DS factors are the main contributors to propagating uncertainties. The uncertainty in interaction (q12) is influenced by the uncertainties in both contributing factors (q1 and q2). The interactions of TEM with PD or Lith and DS with Lith exhibit higher uncertainties. However, the impact of GDP uncertainty diminishes when combined with other factors. Additionally, these three factors of high uncertainty were discretized using the optimal discretization method, indicating that the choice of different discretization schemes in the GDM also introduced uncertainty.

## 5. Conclusions

Uncertainties stemming from measurement errors in soil sampling point locations and concentrations, variations of the exposure variables, and uncertainties of categorial boundaries of influencing factors significantly impact soil heavy metal(loid) pollution and health risk assessment models. To quantitatively evaluate propagations of these uncertainties, we applied the MCS method to the aforementioned models in Hunan Province. The main conclusions are as follows:The heavy metal(loid) pollution in Hunan Province is mainly located in Chenzhou and Hengyang Cities in Hunan Province. Notably, the middle- and high-risk zones largely coincide with the key mining activity planning areas of Hunan Province. Moreover, the findings from the uncertainty analysis revealed a strong correlation between the distribution of areas with high pollution risk and low uncertainty, emphasizing the critical need for prompt actions to address soil pollution in Chenzhou and Hengyang Cities.The carcinogenic and non-carcinogenic risks of children in Hunan Province are higher than those of adults. As and Cr are the primary contributors to health risks, posing probabilities of 25% and 75% for high carcinogenic risks exist in the children population, respectively, while the risks associated with other metals remain within acceptable levels.In the factor detector of GDM, DS, aspect, slope, and PRE emerge as the most influential natural factors affecting the spatial distribution of soil heavy metal(loid) pollution, while DRW, MA, and LULC are identified as the most influential anthropogenic factors. In the interaction detector, the interaction of Lith with other factors (expect PD) is identified as nonlinearly enhanced, while the interactions of other driving factors such as ST, WSL, pH, and SOC are categorized as bivariate enhanced. Regarding uncertainty, TEM, GDP, and DS exhibit the most significant variances, encompassing uncertainties in input data propagation and model accuracy.

While the Nemerow pollution index used in this study is suitable for assessing various heavy metal(loid) elements, the importance of some elements may be overlooked. The comprehensive comparison of the geo-accumulation index and potential ecological risk index can provide a more thorough assessment, reducing the deviation of a single index and improving accuracy. Despite systematically analyzing uncertainties in soil heavy metal(loid) pollution and health risk assessment models, further research should aim to establish a unified indicator for both continuous and categorical model outcomes to better analyze spatial distribution characteristics of uncertainty propagation. The results contribute to land management practices that prevent further accumulation of heavy metal(loid)s in the soil resulting from human activities, thereby reducing pollution and associated health risks.

## Figures and Tables

**Figure 1 toxics-11-01006-f001:**
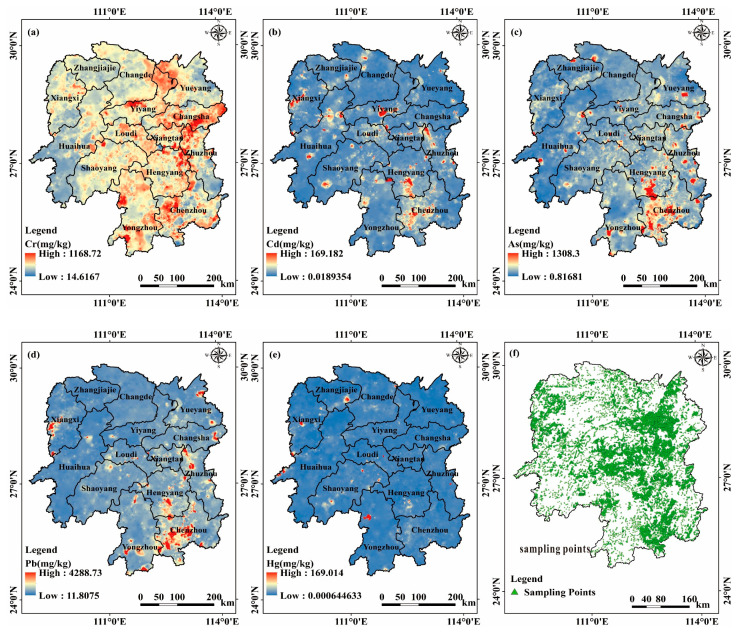
Heavy metal(loid) concentrations and sampling points distribution map: (**a**) Cr, (**b**) Cd, (**c**) As, (**d**) Pb, (**e**) Hg, and (**f**) sampling points.

**Figure 2 toxics-11-01006-f002:**
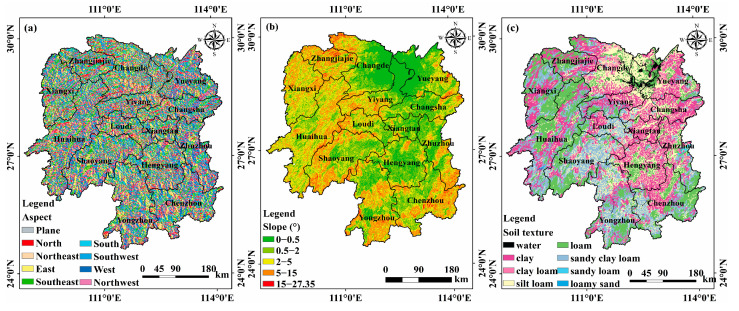
Categorization of natural factors affecting soil heavy metal(loid) pollution in Hunan Province. (**a**) Aspect, (**b**) Slope, (**c**) ST, (**d**) WSL, (**e**) pH, (**f**) DS, (**g**) TEM, (**h**) PRE, and (**i**) SOC.

**Figure 3 toxics-11-01006-f003:**
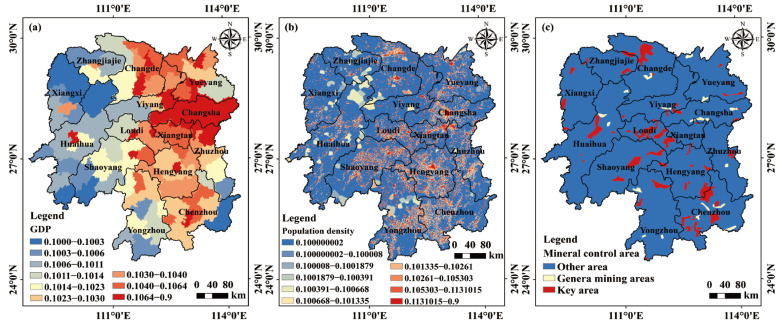
Categorization of anthropogenic factors affecting soil heavy metal(loid) pollution in Hunan Province. (**a**) GDP, (**b**) PD, (**c**) MA, (**d**) LULC, (**e**) DRW, and (**f**) DW.

**Figure 4 toxics-11-01006-f004:**
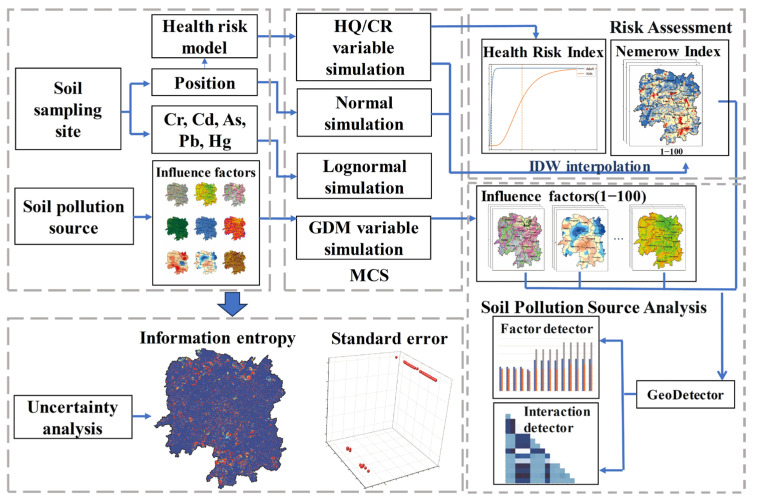
Schematic flowchart of the integrated methodology.

**Figure 5 toxics-11-01006-f005:**
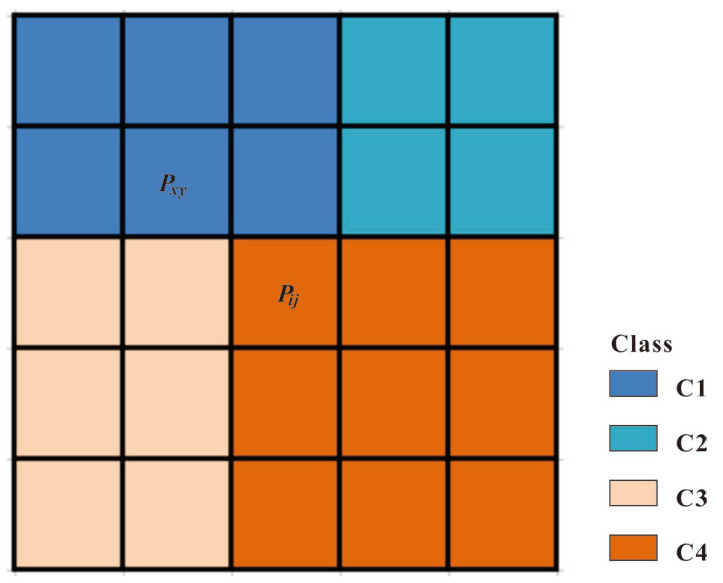
Sample weight diagram of $p_{ij}$ 5 × 5 spatial domain.

**Figure 6 toxics-11-01006-f006:**
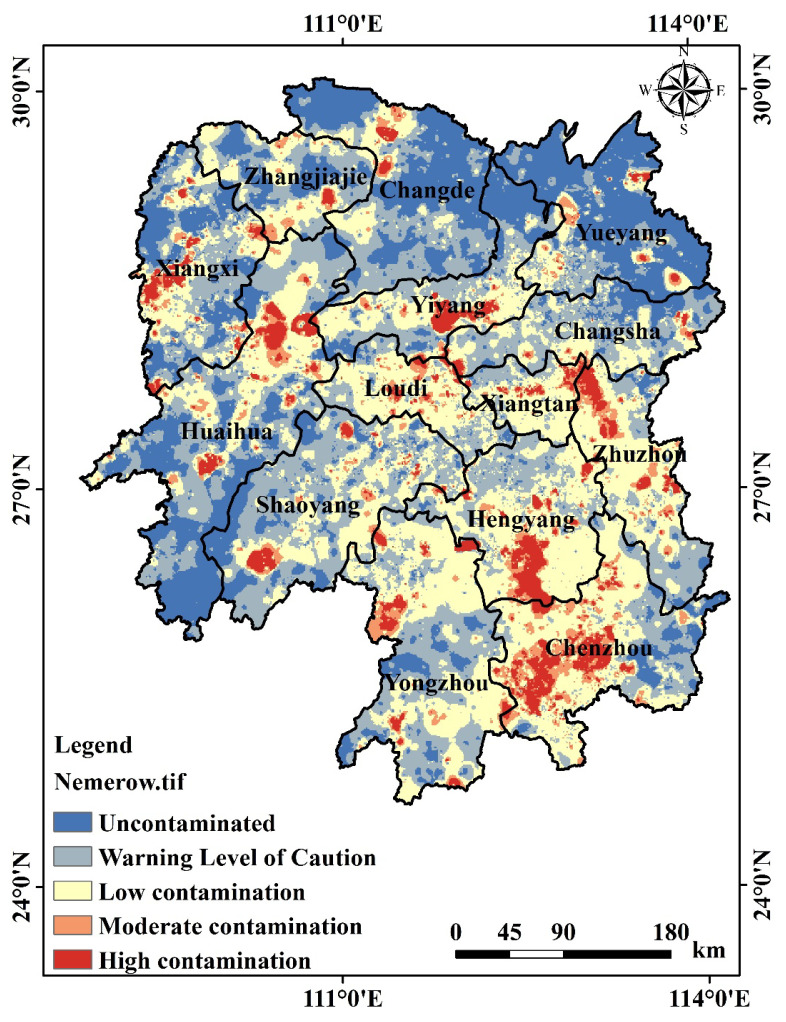
Spatial distribution of Nemerow pollution index $P_{n}$.

**Figure 7 toxics-11-01006-f007:**
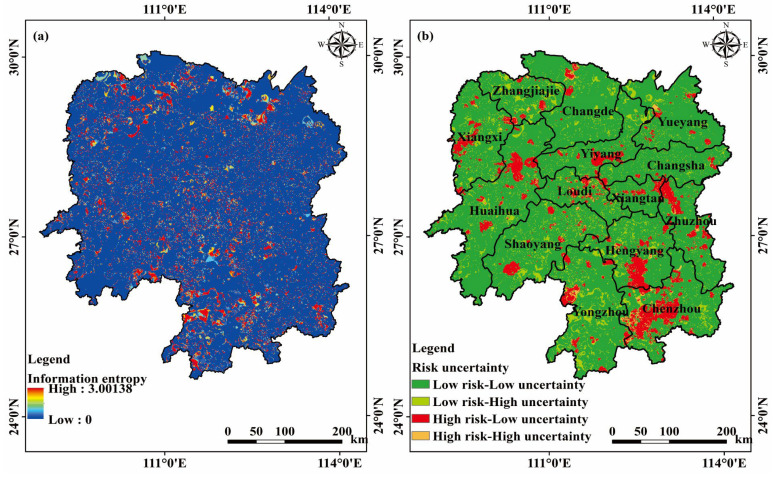
Uncertainty distribution maps of soil heavy metal(loid) pollution risk in Hunan Province: (**a**) information entropy and (**b**) comprehensive control zones.

**Figure 8 toxics-11-01006-f008:**
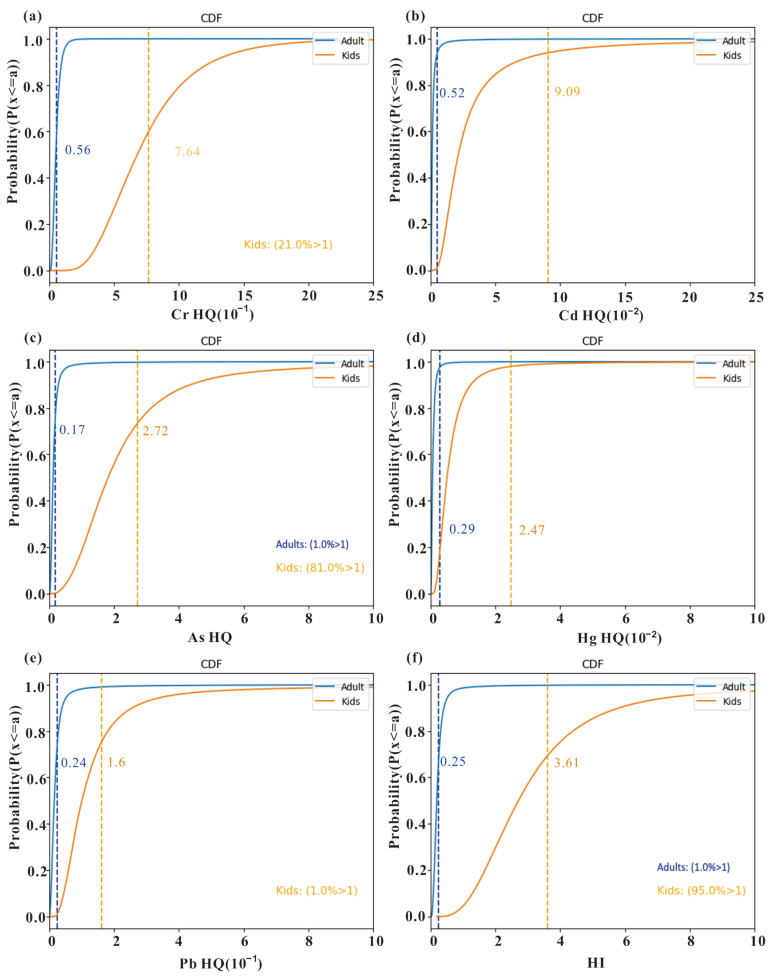
Cumulative probability distribution of non-carcinogenic risk index (HQ) for each heavy metal(loid) of (**a**) Cr, (**b**) Cd, (**c**) As, (**d**) Hg, (**e**) Pb, and (**f**) total non-carcinogenic index (HI). The dashed line represents mean value.

**Figure 9 toxics-11-01006-f009:**
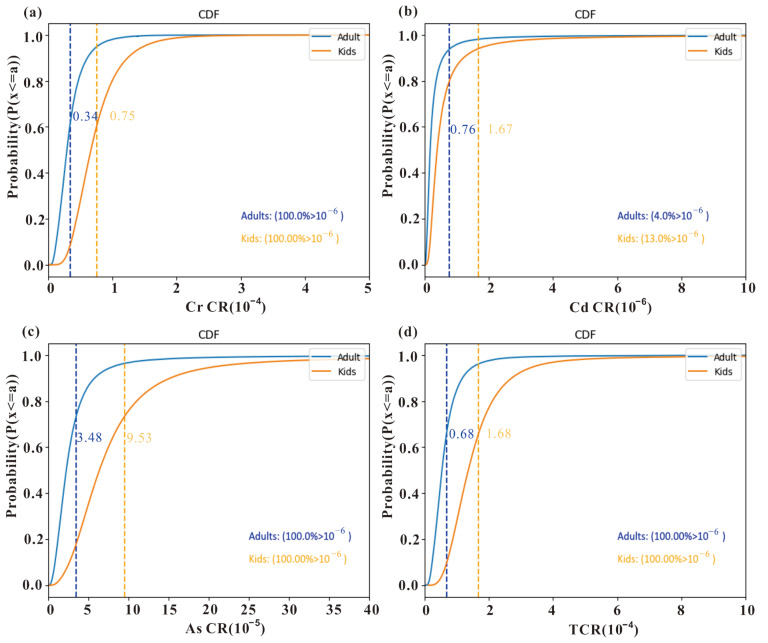
Cumulative probability distribution of carcinogenic risk index (CR) for each heavy metal(loid) of (**a**) Cr, (**b**) Cd, (**c**) As, and (**d**) total carcinogenic index (TCR). The dashed line represents mean value.

**Figure 10 toxics-11-01006-f010:**
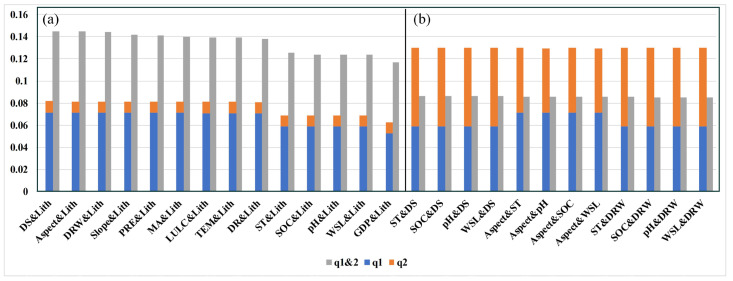
Main interaction types of (**a**) nonlinearly enhanced and (**b**) bivariate enhanced among soil heavy metal(loid)s influencing factors in Hunan Province.

**Figure 11 toxics-11-01006-f011:**
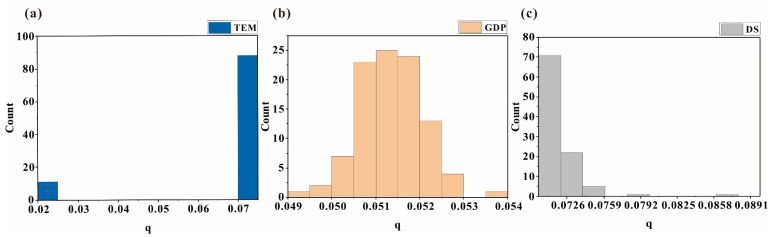
Histogram of q values of the influence factors with the greatest uncertainties: (**a**) TEM, (**b**) GDP, and (**c**) DS.

**Figure 12 toxics-11-01006-f012:**
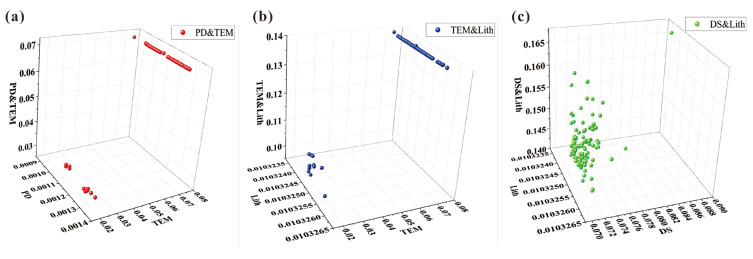
Three-dimensional scatter plot of q values of interaction factors with the greatest uncertainties: (**a**) PD and TEM, (**b**) TEM and Lith, and (**c**) DS and Lith.

**Table 1 toxics-11-01006-t001:** The probabilistic distribution of sampling point attributes and location.

Attribute	Probabilistic Distribution	Parameters LN (50th, 95th)
Cr	Lognormal	LN (9.7, 67.6)
Cd	Lognormal	LN (9.9, 78.0)
As	Lognormal	LN (12.6, 82.7)
Pb	Lognormal	LN (10.1, 71.0)
Hg	Lognormal	LN (10.8, 72.3)
pH	Lognormal	LN (1.9, 4.6)
Position	Normal	N (X,10), N (Y,10)

**Table 2 toxics-11-01006-t002:** Factor Detector q values for sixteen influencing factors of soil heavy metal(loid)s in Hunan Province.

Factors	q
DS	0.07149
DRW	0.07114
Aspect	0.07105
PRE	0.07105
Slope	0.07104
MA	0.07103
LULC	0.07102
TEM	0.07101
DR	0.07050
ST	0.05862
SOC	0.05858
pH	0.05857
WSL	0.05857
GDP	0.05247
Lith	0.01032
PD	0.00129

## Data Availability

The dataset used in this study is not publicly available due to a data privacy agreement with the Hunan Land and Resources Planning Institute. However, it can be obtained from the corresponding author upon reasonable request.

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
