# Peer review of "Uncertainty Evaluation of Soil Heavy Metal(loid) Pollution and Health Risk in Hunan Province: A Geographic Detector with Monte Carlo Simulation"

_toxics, 2023, doi:10.3390/toxics11121006_

Round 1
Reviewer 1 Report
Comments and Suggestions for Authors
1. In the introduction section the authors have to highlight the novelty of the research.
2. The GDM is a very old model, so hoe this can be justified as a novel/new approach.
3. Further the Monte Carlo simulation is also not new. Several research papers has been published from the Hunan Province on this topic so the authors have to justify how their work is different.
4. In Table 2 the SD should be presented along with Mean by + and - sign. The pH is not a metal so include it in the text rather than in the table.
5. Make the conclusion more crisp and to the point. Write about the salient findings.
Comments on the Quality of English LanguageMinor editing is required.
Author Response
Dear Reviewer:
Thanks for your effort to review our manuscript titled "Uncertainty Evaluation of Soil Heavy Metal(loid)s Pollution and Health Risk in Hunan Province: A Geographic Detector with Monte Carlo Simulation", and now we have just revised this manuscript according to your good suggestions. The details are as follows, and all the revisions are done using track changes in Word.
Comments and Suggestions for Authors
- In the introduction section the authors have to highlight the novelty of the research.
Response: Thank you for the suggestion. We have rewritten the last paragraph of Introduction to highlight the novelty of this study. “To address these issues, we employ MCS to jointly analyze uncertainty propagation effects related to sampling point locations, heavy metal(loid) concentrations, exposure variables, and impact factor boundaries on soil pollution assessment, human health risk assessment, and GDM models. In soil pollution assessment, MCS explores uncertainty propagation from sampling positions and heavy metal(loid) concentrations on Nemerow index, quantifying pollution level uncertainty using information entropy. For human health risk assessment, MCS investigates uncertainty propagation from heavy metal(loid) concentrations and exposure variables on health risk index, generating cumulative probability curves. In GDM, MCS probes uncertainty propagation from impact factor category boundaries on Nemerow indices using factor detector and interaction detector models. This study enriches the theoretical methodologies for addressing uncertainty propagation in soil heavy metal(loid)s pollution and health risk assessment models, offering vital support for more precise control of heavy metal(loid)s pollution in Hunan Province.” (L94-118)
- The GDM is a very old model, so hoe this can be justified as a novel/new approach.
Response: Thank you for pointing out this issue. We have rewritten Methods section with subheadings of “Nemerow Pollution Index with MCS” (L280-305), “Health Risk Assessment with MCS” (L335-350), and “Geographic Detector Model with MCS” (L402-442) to highlight the novelty of MCS uncertainty analysis coupled with GDM.
- Further the Monte Carlo simulation is also not new. Several research papers has been published from the Hunan Province on this topic so the authors have to justify how their work is different.
Response: Thank you for the suggestion. We have cited literature on MCS and its application on related topics. “MCS is widely used to assess uncertainties of exposure variables (soil ingestion rate, exposure duration, average body weight, exposed skin area, and skin adherence) to reflect individual differences in human health risk models. For example, Barrio-Parra et al. [1] assessed the effect of variability and uncertainty on all exposure variables. Zhou et al. [2] utilized MCS to evaluate the probabilistic health risks associated with a smelter in Hunan Province. However, measurement errors in sampling point locations and concentrations have been overlooked. Meanwhile, few studies have discussed uncertainty propagation in Nemerow soil pollution evaluation and GDM in conjunction with MCS.” (L85-93)
- In Table 2 the SD should be presented along with Mean by + and - sign. The pH is not a metal so include it in the text rather than in the table.
Response: We have modified Table S2 and revised the description of pH in the text (L172-173).
- Make the conclusion more crisp and to the point. Write about the salient findings.
Response: Thank you for pointing out this issue. We have rewritten the first paragraph of Conclusion to highlight the main findings. “Uncertainties stemming from measurement errors in soil sampling point locations and concentrations, variations of the exposure variables, and uncertainties of categorial boundaries of influencing factors significantly impact soil heavy metal(loid) pollution and health risk assessment models. To quantitatively evaluate propagations of these uncertainties, we applied MCS method to the aforementioned models in Hunan Province.” (L655-660)
Comments on the Quality of English Language
Minor editing is required.
Response: Mr. Syed Yasir Ali Shah (the 3rd author) has revised scientific English writing of this manuscript.
Reference:
- Barrio-Parra, F.; Serrano García, H.; Izquierdo-Díaz, M.; De Miguel, E. Exposure Factors vs. Bioaccessibility in the Soil-and-Dust Ingestion Pathway: A Comparative Assessment of Uncertainties Using MC2D Simulations in an Arsenic Exposure Scenario. Exposure and Health 2023, doi:10.1007/s12403-022-00533-w.
- Zhou, Y.; Jiang, D.; Ding, D.; Wu, Y.; Wei, J.; Kong, L.; Long, T.; Fan, T.; Deng, S. Ecological-health risks assessment and source apportionment of heavy metals in agricultural soils around a super-sized lead-zinc smelter with a long production history, in China. Environmental Pollution 2022, 307, 119487, doi:https://doi.org/10.1016/j.envpol.2022.119487.
Reviewer 2 Report
Comments and Suggestions for Authors
Based on information of the manuscript titled “Uncertainty Evaluation of Soil Heavy Metals Pollution and Health Risk in Hunan Province: A Geographic Detector with Monte Carlo Simulation", the manuscript is interesting to be published in this journal. However, I recommend making minor changes before acceptance.
· Keywords should have key words that are not in the title so that the target audiences have a variety of words to search for the manuscript.
· Arsenic is a metalloid. The term heavy metal(loid) could be considered to include all the elements analyzed.
· The exposure values seem exaggerated. What is the evaluated scenario? Residential, recreational?
· The manuscript has many figures (16) and tables (8), select the most relevant ones, and place the rest in supplementary material.
· Improve the quality of the graphics, for example, figure 16 is not readable.
· Expand the discussion based on other studies where probabilistic analysis is carried out and uncertainty in the evaluations is considered. Recommended documents: https://doi.org/10.1016/j.landurbplan.2019.02.005
https://doi.org/10.1016/j.watres.2016.11.012
,https://doi.org/10.1007/s12403-022-00533-w
https://doi.org/10.1016/j.ecoenv.2021.112629
Author Response
Dear Reviewer:
Thanks for your effort to review our manuscript titled "Uncertainty Evaluation of Soil Heavy Metal(loid)s Pollution and Health Risk in Hunan Province: A Geographic Detector with Monte Carlo Simulation", and now we have just revised this manuscript according to your good suggestions. The details are as follows, and all the revisions are done using track changes in Word.
Comments and Suggestions for Authors
Based on information of the manuscript titled “Uncertainty Evaluation of Soil Heavy Metals Pollution and Health Risk in Hunan Province: A Geographic Detector with Monte Carlo Simulation", the manuscript is interesting to be published in this journal. However, I recommend making minor changes before acceptance.
Response: We appreciate your positive comments and grateful for that. We have addressed your concerns in our manuscript on response to your detailed comments.
- Keywords should have key words that are not in the title so that the target audiences have a variety of words to search for the manuscript.
Response: Thank you for pointing out this issue. We have changed the keywords as “Nemerow index; Non-carcinogenic risk; Carcinogenic risk; Factor detector; Interaction detector; Uncertainty propagation.” (L28-30)
- Arsenic is a metalloid. The term heavy metal(loid) could be considered to include all the elements analyzed.
Response: Thank you for pointing out this issue. We have replaced metal with metal(loid) throughout the manuscript.
- The exposure values seem exaggerated. What is the evaluated scenario? Residential, recreational?
Response: Thank you for pointing out this issue. “The sampling site locations span various agricultural land scenarios: arable land (paddy and dryland), garden land, forest land, and grassland.”(L163-164) We have reconfigured the exposure variable (Figure S5) and recalculated health risks. (L495-509 and L518-527) “Future investigations should explore exposure variable differences in oral ingestion, inhalation via nose and mouth, and dermal contact across arable land, garden land, forest land, and grassland various scenarios.” (L543-545)
- The manuscript has many figures (16) and tables (8), select the most relevant ones, and place the rest in supplementary material.
Response: Thank you for the suggestion. We have moved original Tables 1, 2, 3, 4, 5, and 6 and Figures 2 and 9 from the manuscript to supplementary materials and removed original Figures 1 and 14.
- Improve the quality of the graphics, for example, figure 16 is not readable.
Response: We have improved the clarity of Figure 16 and interpreted it more in the text. (L633-637)
- Expand the discussion based on other studies where probabilistic analysis is carried out and uncertainty in the evaluations is considered. Recommended documents: https://doi.org/10.1016/j.landurbplan.2019.02.005
https://doi.org/10.1016/j.watres.2016.11.012
,https://doi.org/10.1007/s12403-022-00533-w
https://doi.org/10.1016/j.ecoenv.2021.112629
Response: Thank you for providing the excellent references. We have cited these articles in “Uncertainty of Health Risk Assessment”. “Data uncertainty in human health risk assessment arise from environmental variations, population characteristics, and insufficient scientific understanding of parameters and variables. MCS can enhance the quality and quantity of information, thereby reducing parameter uncertainty [1]. Although the Bayesian approach demonstrates precision with small sample sizes [2,3], MCS offers wider applicability and aligns better with the extensive data used in this study.” (L336-341)
Reference
- Barrio-Parra, F.; Serrano García, H.; Izquierdo-Díaz, M.; De Miguel, E. Exposure Factors vs. Bioaccessibility in the Soil-and-Dust Ingestion Pathway: A Comparative Assessment of Uncertainties Using MC2D Simulations in an Arsenic Exposure Scenario. Exposure and Health 2023, doi:10.1007/s12403-022-00533-w.
- Jiménez-Oyola, S.; Chavez, E.; García-Martínez, M.-J.; Ortega, M.F.; Bolonio, D.; Guzmán-Martínez, F.; García-Garizabal, I.; Romero, P. Probabilistic multi-pathway human health risk assessment due to heavy metal(loid)s in a traditional gold mining area in Ecuador. Ecotoxicology and Environmental Safety 2021, 224, 112629, doi:https://doi.org/10.1016/j.ecoenv.2021.112629.
- Barrio-Parra, F.; Izquierdo-Díaz, M.; Dominguez-Castillo, A.; Medina, R.; De Miguel, E. Human-health probabilistic risk assessment: the role of exposure factors in an urban garden scenario. Landscape and Urban Planning 2019, 185, 191-199, doi:https://doi.org/10.1016/j.landurbplan.2019.02.005.
Reviewer 3 Report
Comments and Suggestions for Authors
Dear author,
This paper is very interesting for the reader since it contains important information regarding the pollution of heavy metals (Cr, Cd, As, Hg and Pb) in a great number of soil samples (48811 sampling sites) from a huge area (21.18x104 km2) in Hunan province-China. In addition, the Nemerow index indicating the amount of the pollution, the health risk assessment, and the possible sources from which the pollution may have originated in the soils of this area are examined. I propose that it be published in the journal TOXICS with a small revision – addition regarding the methods of determining the concentrations of these five metals. I would therefore ask the authors to add some sentences after the line 136 of paragraph 2.2.1. of the MS, indicating the extraction methods and analytical determination of the concentrations of the five metals which were used to compile the dataset.
Author Response
Dear Reviewer:
Thanks for your effort to review our manuscript titled "Uncertainty Evaluation of Soil Heavy Metal(loid)s Pollution and Health Risk in Hunan Province: A Geographic Detector with Monte Carlo Simulation", and now we have just revised this manuscript according to your good suggestions. The details are as follows, and all the revisions are done using track changes in Word.
Comments and Suggestions for Authors
Dear author,
This paper is very interesting for the reader since it contains important information regarding the pollution of heavy metals (Cr, Cd, As, Hg and Pb) in a great number of soil samples (48811 sampling sites) from a huge area (21.18x104 km2) in Hunan province-China. In addition, the Nemerow index indicating the amount of the pollution, the health risk assessment, and the possible sources from which the pollution may have originated in the soils of this area are examined. I propose that it be published in the journal TOXICS with a small revision – addition regarding the methods of determining the concentrations of these five metals. I would therefore ask the authors to add some sentences after the line 136 of paragraph 2.2.1. of the MS, indicating the extraction methods and analytical determination of the concentrations of the five metals which were used to compile the dataset.
Response: We appreciate your positive comments and grateful for that. We have added the reference standard for extraction and determination methods. “Multi-point mixture approach was conducted using an "S" shaped or plum blossom coupled points with random sampling techniques, extracting surface soil samples at depths of 0–20 cm. These samples underwent natural air drying in the laboratory, sieving, and subsequent acid digested, adhering to the standards outlined in HJ/T 166-2004 [1]. Determination of heavy metal(loid)s was carried out according to the method specified in GB 15618-2018 [2].” (L165-170).
Reference:
- HJ/T 166-2004. The Technical Specification for soil Environmental monitoring; State Environmental Protection Administration: Beijing,China, 2004.
- GB 15618-2018. Soil Environmental quality–risk control standards for soil contamination of agricultural land; Ministry of Ecology and Environment of the People’s Republic of China Beijing,China, 2018.
Reviewer 4 Report
Comments and Suggestions for Authors
Manuscript titled "Uncertainty Evaluation of Soil Heavy Metals Pollution and Health Risk in Hunan Province: A Geographic Detector with Monte Carlo Simulation" presents an interesting approach to the problem of environmental pollution. The authors present in a broad context a number of factors that may influence the spread of pollution. The uncertainty of the contamination determination is also assessed. A rich mathematical and statistical apparatus was used, which is certainly an advantage of the presented work. I rate the manuscript highly, although it is sometimes written in quite difficult language.
Some remarks:
Please consider moving some figures and tables to an appendix. E.g. Fig. 1 and 2 and Table 1. Thanks to this, the article can gain in clarity.
Figs 13 and 14 say more or less the same thing. In my opinion, Fig. 14 can be omitted without damaging the manuscript.
Fig. 16 in this form is relatively difficult to read - I suggest finding another way to present this data.
The word hydrargyrum (line 136) is obsolete - please replace it with the word mercury.
The link http://www.hngtghy.com/ does not work (or at least it did not work at the time of writing the review).
I also have comments regarding the presentation of some results, which is particularly important considering the fact that the authors discuss "uncertainty" here. Well, the way of rounding the results should be improved. There is no point in giving, for example, the value of kurtosis up to 6 significant figures (Table 2), nor the value of skewness up to 4-5 significant figures. Please review these results and round them appropriately.
When calculating the Nemerov index, the adopted Si values of the "evaluation standard of HM" could be provided for individual metals (line 232), as the quoted GB 15618-2018 standard is difficult to access for readers outside China. This can be provided, for example, in an appendix.
Comments on the Quality of English LanguageEnglish is ok, however the text should be carefully reviewed again before publication due to any ambiguities.
